# Treatment of Severe Swallowing Dysfunction in Systemic Sclerosis with IVIG: Role of Antimuscarinic Antibodies

**DOI:** 10.3390/jcm11226665

**Published:** 2022-11-10

**Authors:** Fabian A. Mendoza, Anthony DiMarino, Sidney Cohen, Christopher Adkins, Shady Abdelbaki, Satish Rattan, Christopher Cao, Susie Denuna-Rivera, Sergio A. Jimenez

**Affiliations:** 1Department of Medicine, Division of Rheumatology, Thomas Jefferson University, Philadelphia, PA 19107, USA; 2Jefferson Institute of Molecular Medicine and Scleroderma Center, Philadelphia, PA 19107, USA; 3Division of Gastroenterology, Department of Medicine, Thomas Jefferson University, Philadelphia, PA 19107, USA

**Keywords:** systemic sclerosis, IVIG, autoantibodies, esophageal dysmotility, treatment

## Abstract

Oropharyngeal and esophageal dysmotility can cause serious clinical complications such as aspiration pneumonia, cachexia, and sarcopenia, with a resulting increase in mortality and disability. The current standard of care for the treatment of SSc-associated swallowing dysfunction is mainly supportive, although severe cases are usually refractory to conventional management. Recent studies have shown that the abnormal production of functional autoantibodies such as anti-cholinergic muscarinic receptor III antibodies may participate in the pathogenesis of SSc-associated gastrointestinal dysmotility and may provide a novel target for therapeutic intervention. We describe two patients with severe and rapid onset of SSc-associated severe swallowing dysfunction and esophageal dysmotility who had failed standard of care therapy, requiring complete enteral and parenteral nutrition. Both patients were positive for the presence of circulating antimuscarinic III receptor antibodies. They were treated with IVIG at a dose of 2 g/Kg/month divided in two consecutive days, for six months. Following IVIG therapy, both patients markedly improved their symptoms as shown by a reduction in their UCLA2.0 score, and achieved an improvement of esophageal motility documented radiologically. Both patients resumed oral feeding and had their feeding tubes removed within the treatment period. None of the patients developed severe adverse events attributable to IVIG, except for low-grade fever during IVIG infusion in one of the cases. These results provide support for the role of functional autoantibodies in the development of SSc-associated gastrointestinal dysfunction.

## 1. Introduction

Systemic Sclerosis (SSc) is a systemic autoimmune disease characterized by prominent fibrosis of the skin and internal organs, vasculopathy, and cellular and humoral abnormalities. These phenomena occur sequentially or simultaneously in different organs causing organ dysfunction and failure. The vast majority of patients with SSc develop esophageal dysmotility that is characterized by ineffective peristalsis and low resting pressure of the lower esophageal sphincter (LES) [1,2,3,4]. Subsequent gastro-esophageal reflux disease (GERD) is common, affecting up to 80% of SSc patients, and is often complicated by the development of esophagitis, esophageal strictures, Barrett’s esophagus, and esophageal cancer [5,6,7,8]. Severe esophageal dysmotility is also strongly associated with the development of oropharyngeal dysmotility and swallowing dysfunction resulting in subsequent complications including aspiration pneumonia, cachexia, and sarcopenia.

The current treatment of esophageal dysmotility includes the use of proton pump inhibitors (PPIs) and promotility agents, as well as procedures such as endoscopic dilation of esophageal strictures [9,10,11]. On the other hand, patients with severe swallowing dysfunction require enteral nutrition or total parenteral nutrition (TPN) to provide adequate nutrition and avoid aspiration of esophageal or gastric content into the lungs [7]. Recently, it has been proposed that the abnormal production of auto-antibodies against muscarinic receptor 3 may contribute to the pathogenesis of GI dysmotility in SSc [12,13,14,15,16]. These auto-antibodies have been found to cause antibody-mediated attenuation of smooth-muscle contraction in animal models [12]. Furthermore, it has been demonstrated in ex vivo studies, that the addition of pooled human Immunoglobulin G (IgG) was able to neutralize the effects of SSc autoantibodies and prevented antimuscarinic receptor-mediated reduction in smooth muscle contraction [15,16]. Based on the results of these in vitro and in vivo studies, Intravenous Immunoglobulins (IVIG) have been proposed as a potential agent for the treatment of SSc-associated GI dysmotility. Inhibition of T cell activation, complement pathway, and the blockade of the Fcγ receptor by IVIG have been also proposed to play a potential role in SSc. Here, we describe our experience with two SSc patients with SSc presenting with acute onset of severe GI dysmotility and swallowing dysfunction that was successfully treated with IVIG.

## 2. Patients and Methods

Two patients fulfilling the 2013 American College of Rheumatology/European League Against Rheumatism (ACR/EULAR) criteria for SSc [17] without myositis, presented with recent onset of severe swallowing dysfunction and esophageal dysmotility, poorly responsive to currently recommended medical therapy that included PPIs, H2 blockers and prokinetic agents. The clinical features of these patients are summarized in Table 1. The patients were treated with IVIG in addition to standard therapy. The patients did not have any history of anaphylaxis or severe adverse reactions to human blood and blood products or any history of a recent deep vein thrombosis or stroke. The potential risks and benefits of IVIG therapy were extensively discussed with the patients, and informed consent was provided by each patient. High-resolution esophageal manometry was performed before initiation of IVIG treatment. The severity of GI symptoms was quantified pre-and post-treatment by the UCLA Scleroderma Clinical Trial Consortium Gastrointestinal Tract Instrument Version 2 (UCLA2.0) [18]. All the data underlying this article are available in the article. Individual patient consent for publication was obtained and patients were provided with a copy of this manuscript.

## 3. Results

### 3.1. Patient 1

A 50-year-old male with a past medical history of moderate mitral valve prolapse and a recent diagnosis of coronary artery disease (CAD) developed symptoms of Raynaud’s phenomenon a few months before the onset of rapidly progressive diffuse skin thickening and progressive dyspnea on exertion. He was found to have a positive ANA (1:1280) with a speckled pattern, a high titer of RNA Polymerase III antibody, and negative anti-centromere and anti-topoisomerase antibodies. The patient was diagnosed with rapidly progressive diffuse cutaneous SSc at another hospital and received treatment with mycophenolate mofetil with very poor gastrointestinal tolerance.

At the initial evaluation at Thomas Jefferson University Scleroderma Center, the patient had seven months since the first noticeable occurrence of skin thickening and complained of dyspnea on exertion, painful joint contractures. The patient also developed and severe dysphagia and swallowing dysfunction, nausea, regurgitation, and frequent post-prandial vomiting causing a very low caloric intake of less than 400 calories/day leading to severe weight loss. These symptoms persisted despite of use of proton pump inhibitors (PPIs). Prokinetics were not tolerated. Physical examination disclosed severe weight loss (greater than 100 pounds or approximately 50% of his normal weight), marked diffuse skin thickening with proximal limb and trunk involvement, and a modified Rodnan Skin Score (mRSS) of 25. Multiple joint contractures and tendon friction rubs were evident on musculoskeletal exam. Auscultation of the chest disclosed bibasilar crackles and a mitral valve click sound. The abdomen was distended but not tender with normal bowel sounds. Additional testing showed severe lung restriction at Pulmonary Function Tests (FVC 25% of the predicted value with unobtainable DLCO despite repetitive testing). A high-resolution computerized tomography of the chest showed reticulonodular opacities with few bronchiectasis and a patulous esophagus. An echocardiogram demonstrated a low ejection fraction (EF 35%) without regional wall motion abnormalities, nor evidence of pulmonary hypertension. Right and left heart catheterization displayed normal pulmonary artery pressures and resistance, and CAD. Laboratory tests showed anemia, elevated inflammatory markers (CRP, ESR, and Ferritin), normal creatine kinase (39 IU/L), and a negative myositis autoantibody panel. Furthermore, electromyography (EMG) did not show evidence of myopathy or neuropathy. Owing to the severity of GI dysmotility, an enzyme-linked immune-sorbent assay for anti-muscarinic receptor 3 antibodies (Aviva, San Diego, CA, USA) was performed finding positive results with a concentration calculated at 1.86 ng/mL (test range 0.312–20 ng/mL). High-resolution esophageal manometry demonstrated absent contractility of the esophagus without any measurable contraction of the esophageal body during any swallow and normal resting pressure of the LES (27.3 mmHg). Upper esophageal sphincter (UES) resting pressure and relaxation were normal as well pharyngeal peristalsis was preserved. In addition, upper endoscopy did not show macroscopic or microscopic evidence of eosinophilic esophagitis.

The patient failed to improve swallowing or recover esophageal functionality following esophageal sphincter dilation and was not able to tolerate institution of oral intake. He was started on TPN and gastric-jejunal enteral feeding. The patient was then treated with parenteral cyclophosphamide (750 mg/m^2^) monthly administration for SSc-associated interstitial lung disease for six months in addition to a small dose of prednisone (5 mg/d) that was started prior to the onset of GI symptoms for the control of joint pain. Owing to the severity of the GI involvement and the presence of antimuscarinic antibodies in the serum and following informed consent the patient was treated with IVIG (1 g/Kg/dose) for 2 consecutive days every month for 5 months (see Figure 1A) of a total of 6 planned infusions (the patient declined the last infusion given his remarkable improvement) with no adverse events. Following six months of therapy with cyclophosphamide and IVIG, the patient improved the skin sclerotic changes with a reduction of mRSS to 15 and improvement of lung volumes. Furthermore, a combination of TPN, iron infusions, and low-volume tube feeding was used to maintain appropriate nutrition. With clinical improvement the patient was progressively changed to oral intake, achieving complete oral caloric requirements 6 months following initiation of IVIG. His UCLA 2.0 GI score markedly improved from 1.75 to 0.175. He was switched from cyclophosphamide to oral mycophenolate with good GI tolerance and stopped IVIG after 6 months without recurrence of symptoms. He regained 80 pounds of weight and he was able to re-initiate his physical exercising activities. However, high-resolution esophageal manometry following treatment failed to demonstrate any evidence of esophageal peristalsis improvement.

### 3.2. Patient 2

A 72-year-old female with a past medical history of iron deficiency anemia secondary to intestinal angiodysplasia, Raynaud’s phenomenon for 10 years, and GERD for 13 years treated with PPIs, was hospitalized with a sudden onset of fever, shortness of breath, cough, severe dysphagia, and possible aspiration pneumonia following a plastic surgical procedure (Rhytidectomy). She was initially treated with parenteral antibiotics and hydration. Following antibiotic therapy, the patient became afebrile but complained of dyspnea on exertion and severe dysphagia. A physical exam was remarkable for the inability to swallow her saliva, the presence of crackles in the right lung base and a normal abdominal exam. She had sclerodactyly and facial skin thickening with a mRSS of 3. Few telangiectasias in palms were noticed. The patient lacked sicca signs or symptoms suggestive of Sjogren’s syndrome. Laboratory tests showed mild anemia with a hemoglobin level of 9 g/dL and a positive serum ANA test at a 1:640 titer with an anti-centromere antibody pattern. Serum inflammatory markers and muscle enzymes and creatinine kinase level were normal. Myositis antibody panel was negative as well as no evident neuromuscular abnormality was found at EMG. The presence of anti-muscarinic receptor 3 antibodies was demonstrated by enzyme-linked immune-sorbent assay (Aviva, San Diego, CA, USA) and its concentration was calculated at 1.57 ng/mL (range 0.312–20 ng/mL). A chest computed tomography disclosed ground glass opacities in the right lower lung field and endo-bronchial material in airways compatible with aspiration pneumonia. An echocardiogram did not show evidence of increased pulmonary artery pressure. High-resolution esophageal manometry demonstrated absent contractility of the esophagus without any scorable contraction during any swallow. Upper esophageal sphincter (UES) resting pressure was low (20 mmHg) with normal pharyngeal peristalsis. Upper endoscopy did not show evidence of eosinophilic esophagitis. A fluoroscopic swallow study displayed mildly delayed oral transit and laryngeal penetration. Pulmonary function tests performed following pneumonia recovery showed normal lung volumes (FVC 97% of predicted) and a mild decline in diffusion capacity (DLCO 60% of predicted).

A gastric tube was placed and enteral nutrition was initiated. Following informed consent, she was started on IVIG 1 g/kg/day for 2 consecutive days every month with no adverse events except for low-grade fever post-administration. Over the subsequent several months, the patient regained her ability to swallow and successfully transitioned from enteral nutrition to oral nutrition 4 months following IVIG therapy initiation. Timing of these clinical events and therapy are summarized in Figure 1B. Her UCLA 2.0 GI score improved from 0.88 to 0.075. She completed 6 months IVIG infusion therapy. High-resolution esophageal manometry failed to demonstrate improvement of esophageal peristalsis, but a video swallow study showed marked improvement and normalization in the swallowing function with resolution of tracheal aspiration.

## 4. Discussion

Severe swallowing dysfunction in SSc patients is not only associated with increased mortality but also is frequently refractory to conventional therapy, requiring prolonged enteral and parenteral nutrition [11]. Recent studies have shown that functional antimuscarinic M3-R receptor autoantibodies present in the serum of certain SSc and Sjogren’s disease patients are capable of blocking gastrointestinal smooth muscle contraction in vivo and this can be reversed with pooled human IgG by neutralizing these circulating autoantibodies [13,14,15,16]. The hypothesis of imbalance of the stimulatory-inhibitory balance of the enteric nervous system caused by M3-R blocking antibodies is illustrated in Figure 2. Furthermore, it has been suggested that immune-mediated mechanisms may play a role in the development of a wider spectrum of gastrointestinal dysmotility in idiopathic gastroparesis and paraneoplastic disorders, and the term autoimmune-mediated gastrointestinal dysmotility (AGID) has been coined to refer to gastrointestinal dysfunction caused by functional auto-antibodies, disorders of Cajal cells, and local or systemic inflammatory disorders [19,20]. Following this hypothesis, illustrated in Figure 2, a series of patients with refractory and idiopathic gastroparesis have been treated with IVIG with highly encouraging results, showing improvement in nausea, vomiting, early satiety, and abdominal pain [19]. In addition, in another recent retrospective review, patients with generalized autoimmune dysautonomia, evidence of neuro-inflammation, and gastroparesis resistant to drug and gastric electrical stimulation also showed improved symptoms following the use of IVIG [20].

In SSc overlapping with polymyositis, the use of IVIG has shown improved GERD and Gastrointestinal symptoms [21,22,23]. However, the possibility of GI symptoms improvement mediated by general improvement of the muscle function due to decreased muscle inflammation was a significant confounding factor. Intriguingly, after more than 12 months of follow up the patients described in this report did not have any recurrence of symptoms nor required a new IVIG cycle. This was in marked contrast to the longer treatment required in patients with overlapping myositis [22].

Based on the growing body of evidence of functional antibodies mediating SSc-associated severe esophageal dysmotility, we chose to treat the cases described in this report with IVIG. Both patients presented here had inability to swallow, esophageal aperistalsis, and anti-muscarinic antibodies. They did not have overlapping myositis. After IVIG, both patients recovered swallowing function, regaining weight, and allowing termination of parenteral and enteral nutrition. The results of this study demonstrate marked clinical improvement in swallowing function in both patients following IVIG. Both of the cases also demonstrated dramatic symptom improvement as shown by the UCLA2.0 score and were able to normalize their oral intake. The use of high-resolution manometry has been a valuable clinical tool to diagnose esophageal involvement in systemic sclerosis but has not been shown to be of clinical value in testing response to a novel therapy such as IVIG. Neither patient showed improvement in esophageal peristalsis after IVIG despite marked clinical improvement in swallowing function. This is a limitation of this clinical report and will require further study. Improvement in peristalsis by high-resolution manometry may require a longer observation period or these patients may have already displayed permanent esophageal neuromuscular damage. It is also possible that subclinical improvement of motility could be enough to improve swallowing function.

It is important to emphasize that the selection of patients with very early and acute GI severe dysmotility, likely occurring prior to structural myogenic damage, can also explain the dramatic improvement of function observed in the patients described here. In addition, none of these patients had associated myositis, a condition associated with GI dysmotility and swallowing dysfunction that can be improved by the use of IVIG. We fully recognize that the concomitant use of cyclophosphamide immunosuppression for the rapidly progressive skin and ILD may be considered a confounding factor in the first patient but rapid improvement of swallowing function is not commonly observed neither in SSc patients treated with cyclophosphamide nor as a natural evolution of the disease. Furthermore, in a large inception cohort of SSc Canadian and Australian patients, the risk of developing severe gastrointestinal involvement was not modified by exposure to immunosuppression [24]. Although no specific analysis of the cases with swallowing dysfunction was made in this study, the data support our clinical observations. We are also very aware that pilot studies and case series with small number of patients are not intended to evaluate the efficacy or safety of a specific therapeutic intervention. However, we are convinced that they are useful to help understanding the pathophysiology and management of rare diseases and to support the initiation of larger controlled studies. This report, with a markedly positive outcome, encourages the performance of more extensive and randomized controlled trials for evaluating the potentially beneficial effects of IVIG administration in SSc patients with new onset of severe swallowing dysfunction. In addition, in this pilot experience, high-resolution esophageal manometry proved not to be useful to quantify the clinically significant improvement, suggesting the need to explore other methods with greater sensitivity to quantify the change in esophageal function.

## Figures and Tables

**Figure 1 jcm-11-06665-f001:**
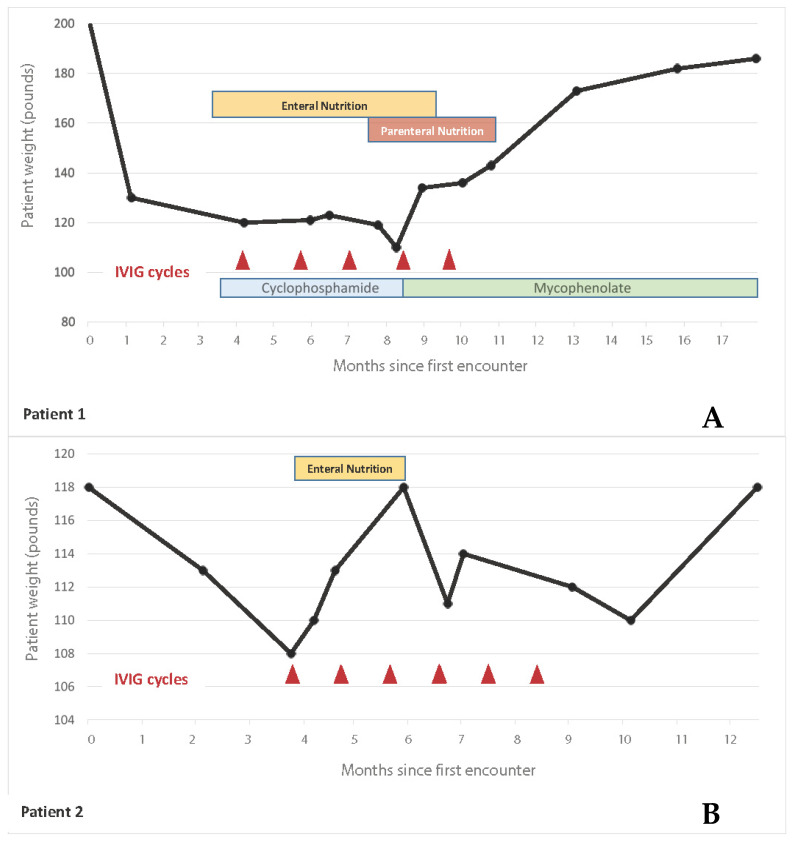
**Evolution of weight, nutritional therapy, additional immunosuppressive therapy, and IVIG over one year.** This figure shows patients’ weight trend (black line) over time in relation to IVIG cycles (red triangles) and adjunctive immunosuppression therapy (in patient 1) (**A**). Both patients were able to wean off nutritional support (enteral and parenteral nutrition in patient 1 and enteral nutrition in patient 2) (**B**).

**Figure 2 jcm-11-06665-f002:**
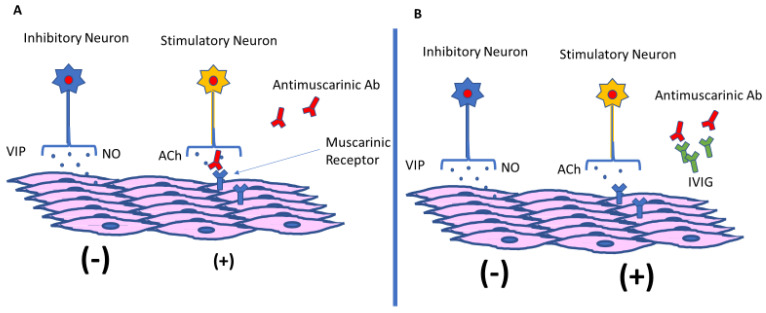
**Hypothesized effect on IVIG in restoring GI motility in SSc patients.** Motor neurons, stablish the connection of the enteric nervous system with the muscles of the GI tract. Excitatory and inhibitory neural signal allow proper GI motility. In SSc dysmotility, antibodies against ACh-muscarine receptors block stimulatory pathways, causing a predominance of the inhibitory signaling over the GI musculature. This imbalance is expressed as dysmotility with lack of proper peristalsis (**A**). IVIG, blocks the abnormal auto-antibodies, helping to restore the altered equilibrium and to recover normal motility (**B**). ACh: acetylcholine, VIP: vasoactive intestinal peptide; NO: nitric oxide.

**Table 1 jcm-11-06665-t001:** Clinical Features of SSc patients prior to IVIG treatment.

Clinical Features	Patient 1	Patient 2
**1-Age/Sex**	50/Male	72/Female
**2-Skin Thickening distribution/mRSS**	Diffuse, involving trunk and proximal upper and lower extremities/mRSS of 25	Limited, involving fingers and face/mRSS of 3
**3-Raynaud’s**	Started 8 months prior to presentation	Started 10 years prior to presentation
**4-Calcinosis**	Not present	Not present
**5-Interstitial lung disease**	Nonspecific interstitial pneumonia on HRCT	Not present on HRCT
**6-Pulmonary arterial hypertension**	Not present per ECHO, right heart catheterization	Not present per ECHO
**7-ANA titer and pattern**	1:1280, speckled pattern	1:640, anti-centromere pattern
**8-Myositis Antibody panel ***	Negative	Negative
**9-Scleroderma specific antibodies** **(a)** **Anti Topoisomerase Ab** **(b)** **Anti RNA Polymerase III Ab** **(c)** **Anti Centromere Ab**	(a) Negative (b) Positive (c) Negative	(a) Negative (b) Negative (c) Positive
**10-Anti-muscarinic receptor 3 Ab**	Positive at 1.86 ng/mL (test range 0.312–20 ng/mL)	Positive at 1.57 ng/mL (test range 0.312–20 ng/mL)

**Abbreviations:** mRSS: modified Rodnan Skin Score, HRCT: High Resolution Computed Tomography, ANA: Anti Nuclear Antibody. * Anti-PM/Sci-100 Ab; Anti-MDA5 Ab (CADM-140); Anti-NXP-2 Ab; Anti-TIF-1gamma Ab; Anti-SAE1 Ab, IgG; Anti-SRP Ab; Anti-U3 RNP; Anti-U2 RNP Ab; Anti-Mi-2 Ab; Anti-PL-7 Ab; Anti-PL-12 Ab; Anti-EJ Ab; Anti-OJ Ab; Anti-Ku Ab; Anti-Jo-1 Ab; Anti-U1 RNP Ab; Anti-SS-A 52kD Ab.

## Data Availability

The data presented in this study are available in the article.

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
