# Peer review of "Treatment of Severe Swallowing Dysfunction in Systemic Sclerosis with IVIG: Role of Antimuscarinic Antibodies"

_jcm, 2022, doi:10.3390/jcm11226665_

Round 1

Reviewer 1 Report

The authors described 2 cases of severe dysphagia occurring in systematic sclerosis. The 2 patients also have muscarinic-3 acetylcholine receptor autoantibody. The authors report their experience of the use of IVIG in these 2 patients who had failed conventional treatments and a priori without associated myositis. The clinical cases are detailed, but missing information had to be provided. 

Major Comments:

Several points significantly narrow the scope of this work:

a)     Severe dysphagia is exceptional in systemic sclerosis, especially in the absence of concomitant malabsorption syndrome and/or pseudo-obstruction and/or other signs of severe digestive disorders. In the literature provided by the authors, there are no similar cases, except in cases of association with polymyositis. 

b)    In the case 1, the concomitant treatment by CYC significantly limits the assessment of the benefit of IVIG in this patient. Any conclusion is very difficult to substantiate. 

c)     In both cases, the authors should specify whether an ENMG was performed in search of asymptomatic myogenic involvement, especially since they cite a study on overlapping syndromes with polymyositis where the existence of specific antibody and rhabdomyolysis may be inconsistent.

d)    In the case 2, the patient complained of swallowing disorders while there was no damage to the oropharyngeal muscles. The authors should specify whether there was an overlap with Sjogren's syndrome.

e)    The authors should mention whether esophageal biopsies were performed to rule out eosinophilic esophagitis which may also be a cause of severe dysphagia.

f)      The authors should mention if other treatments were associated with IVIG: PPI, prokinetics, antibiotics for microbial overgrowth, …? For example, in case 1, the rapid onset of SSc and the early initiation of CYC and IVIG raises the question of the hindsight taken on the other treatments tested mentioned in the introduction?

Minor Comments: 

-       Although widely mentioned in the discussion, the authors should stipulate in case 2 the results of the muscle panel antibodies. 

-       Authors should mention whether steroid was used, including at low dosage (due to the SSc and/or anti-RNApolIII Ab).

-       It’s not clear if the case 1 had swallowing dysfunction or only dysphagia? 

-        The authors should advocate exploring the prevalence of muscarinic-3 acetylcholine receptor autoantibodies in systemic sclerosis in meaningful samples to get an idea of ​​their potential implications.

-   The author should consider that the improvement of digestive involvement could be unrelated to the blockage of muscarinic-3 acetylcholine receptor autoantibodies but to the anti-inflammatory/fibrotic effects of IVIG ?

Reviewer 2 Report

Thank you for the oppertunity to review this interesting and sound case report. Your observations are of great interest pointing out a gap of treatment in systemic sclerosis patient with severe swallowing dysfunction. The role of antimuscarinic antibodies is of great interest to all Ssc experts around the worlde.

I have no further comments. I have all information needed and discussion takes relevant perspectives into consideration. And my congratulations - The perspectives of your observations hopefully will end up helping many Ssc patients in the future.

Reviewer 3 Report

The authors present two patients with GI manifestations of SSc and their response to IVIG. The cases are clearly presented. These reports are of scientific interest. I have a few questions

- Is it possible to characterise the GI involvements further - with respect also to lower GI tract?

- Could you further present what happened after IVIG was cessated. How long follow-up - did GI symptoms relapse?

- In patient A, do you think the initial weight loss was due solely to GI dysmotility - is it not plausible that initial weight loss was multifactorial, associated with the systemic nature of SSc? Could this be discussed?

- Would you like to present the subgroup-data on the GIT 2.0 used?

- Could you discuss further to what degree the reported data could be explained also by the natural history of the disease?

- If IVIG was delivered without complications (e.g. vein thrombosis) this could be mentioned.

- Has other SSc patients at your clinic been treated with IVIG because of GI disease? If not, it could maybe be stated more clearly?

Round 2

Reviewer 1 Report

I thank the authors for their responses. No additional comment.